# Good Long-Term Prognosis of Lupus Nephritis in the High-Income Afro-Caribbean Population of Martinique with Free Access to Healthcare

**DOI:** 10.3390/jcm11164860

**Published:** 2022-08-19

**Authors:** Benoit Suzon, Fabienne Louis-Sidney, Cédric Aglaé, Kim Henry, Cécile Bagoée, Sophie Wolff, Florence Moinet, Violaine Emal-Aglaé, Katlyne Polomat, Michel DeBandt, Christophe Deligny, Aymeric Couturier

**Affiliations:** 1Department of Internal Medicine, Martinique University Hospital, CEDEX CS, 90632 Fort-de-France, Martinique, France; 2Department of Rheumatology, Martinique University Hospital, CEDEX CS, 90632 Fort-de-France, Martinique, France; 3Department of Nephrology, Martinique University Hospital, CEDEX CS, 90632 Fort-de-France, Martinique, France

**Keywords:** Afro-Caribbean, systemic lupus, lupus nephritis, long-term prognosis, end-stage renal disease, mortality

## Abstract

Lupus nephritis (LN) has been described as having worse survival and renal outcomes in African-descent patients than Caucasians. We aimed to provide long-term population-based data in an Afro-descendant cohort of LN with high income and easy and free access to specialized healthcare. Study design: We performed a retrospective population-based analysis using data from 2002–2015 of 1140 renal biopsies at the University Hospital of Martinique (French West Indies). All systemic lupus erythematosus patients with a diagnosis of LN followed for at least 12 months in Martinique or who died during this period were included. Results: A total of 89 patients were included, of whom 68 (76.4%) had proliferative (class III or IV), 17 (19.1%) had membranous (class V), and 4 (4.5%) had class I or II lupus nephritis according to the ISN/RPS classification. At a mean follow-up of 118.3 months, 51.7% of patients were still in remission. The rates of end-stage renal disease were 13.5%, 19.1%, and 21.3% at 10, 15, and 20 years of follow-up, respectively, and mortality rates were 4.5%, 5.6%, and 7.9% at 10, 15, and 20 years of follow-up, respectively. Conclusions: The good survival of our Afro-descendant LN patients, similar to that observed in Caucasians, shades the burden of ethnicity but rather emphasizes and reinforces the importance of optimizing all modifiable factors associated with poor outcome, especially socioeconomics.

## 1. Introduction

Systemic lupus erythematosus (SLE) is an autoimmune disease with severe renal involvement. SLE and lupus nephritis (LN) are described as more prevalent and severe in Afro-Caribbean and Afro-American populations [1,2] with a five-year survival of less than 70% [2,3,4] compared with more than 88% in the Caucasian population [5,6,7]. However, there are discrepancies about whether race is an independent risk factor, with some studies emphasizing the critical role of healthcare access for minorities [8,9,10]. Martinique is a French Caribbean island whose population is more than 90% Afro-descendant and has full and free access to medical care [1,11,12]. A previous study in Martinique estimated the incidence of SLE to be 4.7/100,000 population and showed ten-year overall survival rates similar to those observed in the Caucasian population, suggesting that prognosis in the Afro-Caribbean population is not related to ethnicity [1]. In the present study, we sought to assess the long-term prognosis of lupus nephritis in a population with free and easy access to specialized healthcare.

## 2. Materials and Methods

We practiced a monocentric hospital-based retrospective study of renal biopsies analyzed from 1 January 2002 to 16 September 2015 in the pathology unit of Fort de France University Hospital. Inclusion criterion was histological LN diagnosis according to ISN/RPS classification in SLE patients fulfilling ACR 1997 criteria, with at least 12 months of follow-up. Samples first analyzed with WHO classification were converted to International Society of Nephrology/Renal Pathology Society (ISN/RPS) classification [13]. Patients with less than 12 months of follow-up were excluded, excepted for those who died during this period.

At baseline (first LN) and during follow-up, we collected data such as age, sex, date of SLE diagnosis, SLE treatment, date and histopathological class of LN occurrence, and number of LN flares; biological data such as serum creatinine, urine protein/creatinine ratio hematuria, and detection of anti-nuclear, anti-extractable nuclear antigen (ENA) and anti-phospholipid (aPL) auto-antibodies. Medical history and treatment of high blood pressure (defined as uncontrolled if >130/80 mmHg during 2 consecutive visits) were also collected.

### 2.1. Definitions

In SLE patients fulfilling ACR 1997 criteria, LN was suspected with proteinuria > 0.5 g/24 h or a urine protein/creatinine ratio > 0.5 g/g on two consecutive samples, with or without hematuria (>10,000 red blood cells (RBCs)/mL). LN suspicion was confirmed and classified by kidney histopathological analysis according to ISN/RPS classification. Time to remission was defined by the time from LN diagnosis to complete or partial remission, with or without treatment. Immunosuppressive therapy initiated for LN purposes was referred to as “induction” or “maintenance therapy”. Non-adherence to treatment was not systematically evaluated but was considered when stopping of steroids or immunosuppressant drugs was self-reported. Complete response (CR) was defined if proteinuria was ≤0.5 g/24 h or urine protein/creatinine ratio < 50 mg/mmol, and creatinine no greater than 15% above baseline. Partial response (PR) was defined if proteinuria was ≤3 g/24 h or urine protein/creatinine ratio <300 mg/mmol, with a reduction > 50% from baseline, and creatinine no greater than 15% above baseline [14]. Relapse was defined by the serum level of creatinine increasing >25% above nadir, persistent for more than a month and/or proteinuria increasing on 2 consecutive urinary samples: >1 g/24 h if the patient was in CR, and doubling of proteinuria or >2 g/24 h in case of PR [3]. Relapse duration was considered until re-remission. End-stage renal disease (ESRD) was defined when renal replacement therapy was necessary more than 3 consecutive months.

In case of patients lost to follow-up or missing values, we considered the patient’s status unchanged until a testified new condition. Serum level creatinine, ESRD, and vital status were reviewed for each patient lost to follow-up. If the patient could not be reached, the family physician or other treating physician was contacted by telephone or mail.

Finally, we separated a poor prognosis group for LN, defined by ESRD or death, and compared them to other patients.

### 2.2. Statistics and Ethics

Quantitative data are expressed as mean with standard deviation and qualitative data in unit and percentage. Comparison of the data between groups was performed by Fisher’s exact test or student’s t-test as appropriate with a significant value if *p* < 0.05. Survival curves were computed using the Kaplan Meier method. Analyses were conducted using Prism-GraphPad software. This study was declared to the Commission Nationale Informatique et Libertés (CNIL) with the registration number 1899602v0.

## 3. Results

A total of 89 Afro-Caribbean patients with LN fulfilling ACR 1997 criteria were included, with a mean follow-up of 119.3 ± 73.3 months. Baseline characteristics are given in Table 1.

Among all LN (*n* = 89), 68 (76.4%) were proliferative. Mean SLEDAI score was 17.11 ± 5.73 (*n* = 89). Although the overall time to loss to follow-up was 4.2% of the total follow-up time, the renal and vital status of every single patient were known at the end of the study. Mean time from SLE diagnosis to first LN was 39.7 ± 60.8 months. Hydroxychloroquine (HCQ) was taken for at least 24 months by 74% of patients. Induction therapy consisted of intravenous cyclophosphamide (CYC) in 54 (60.7%) and mycophenolate in 20 (22.5%) cases. Twelve patients received inadequate induction therapy (azathioprine or steroids alone). Three patients did not receive induction therapy: one because of concomitant treatment for lymphoma and the other two for no apparent reason. Regarding maintenance therapy, 64 patients (70.8%) received mycophenolate, 4 (4.5%) received CYC, and 4 (4.5%) received AZA. Other patients received either steroids, ciclosporine, methotrexate, or no treatment. Furthermore, 13 out of 17 (76.47%) initial isolated membranous LN secondarily evolved to a proliferative LN (class IV + V; *n* = 10 and class III + V; *n* = 3) at a mean time of 95 ± 41.2 months. Finally, one of three class I LN progressed to class V and the single class II had a proliferative recurrence (IV + V).

### Evolution and Long-Term Prognosis

Seventy patients (78.7%) went into remission in a mean of 18.7 months [0.9–115]. CR and PR were achieved in 49.5% and 29.2%, respectively. Among responders, 33 (47.1%) never relapsed. Among relapsers, 21 patients (23.6%) presented one, 10 patients presented two (11.2%), and 6 presented patients (6.7%) three renal flares. Evolution features at 1, 5, 10, 15, and 20 years are detailed in Figure 1 and Figure 2. Renal and global survival analysis are given in Figure 3.

Mean estimated renal and global survival were 18.67 ± 1.42 and 20.61 ± 1.06 years, respectively. At the end of follow-up, 19 (21.3%) patients were in ESRD: 8 had a kidney transplant and 11 were treated with hemodialysis, with a mean time from LN to hemodialysis initiation of 92 ± 60 months. Eight (9%) patients died: three of infectious origin (severe chikungunya, bilateral pneumonia, acute pyelonephritis), one of hemorrhage following abdominal surgery, one of probable massive pulmonary embolism, one of heart failure, and two of unknown cause. Poor outcomes were associated with time to remission (27.5 ± 33.4 versus 87.1 ± 58.9 months, *p* < 0.0001) and uncontrolled hypertension (16% vs. 53.8%, *p* = 0.0002).

## 4. Discussion

Afro-descendant or African race or ethnicity is cited as an independent risk factor for adverse outcomes in LN. This finding is not shared by most authors who have highlighted the unfavorable socioeconomic factors of Afro-descendants compared with Caucasians, and their consequences [15,16,17,18,19]. Most populations of African descent worldwide, even in developed countries, face barriers to care, poor socioeconomic conditions, and poverty, which have a negative impact on the prognosis of SLE and LN [20,21,22]. On the contrary, our population cohort of African-descent is unique because of the easy and completely free access to healthcare and specialized care providers (national reference center dedicated to SLE). Despite a high activity score and high proportion of proliferative LN, we reported good long-term outcomes of our Afro-Caribbean population, similar to that observed in Caucasian patients [5,6,7].

To compare survival and ESRD in studies coming from tertiary centers or population based can be hazardous but all available data on African-descent patients go on the same way. To date, survival in observational studies concerning LN African-descent patients remains consistently low: 69% at 5 years in an Afro-Caribbean population from the UK [23], 59% at 10 years for an African-American population [4]; and 91 and 59%, 93 and 68%, 90.9 and 60.7% at 1 and 5 years in Curacao [2], Barbados [24], and Jamaica [25], respectively. Mortality was also superior in African-American LN patients than Caucasians with Medicaid or Medicare insurance [26]. In the same way, ESRD has been consistently found more frequently in studies including African-descent patients in [27,28,29] and outside the United States [23] compared to Caucasian populations. Some studies have compared the long-term outcomes of lupus nephritis between patients of African origin and Caucasians. For example, one-, five-, and ten-year survival of African-descent versus Caucasian LN patients was 95 vs. 94%, 71 vs. 85%, and 59 vs. 81%, respectively. At the same end points, renal survival was 85 vs. 91%, 50 vs. 74%, and 38 vs. 68%, respectively [4]. Another study reported a five-year renal survival in African-descendants of 57% compared to 94.5% in Caucasians [27]. In a largely Caucasian population (76 Caucasians, 8 Afro-descendants, and 6 Asians), who had received recent immunosuppressive treatments (Eurolupus protocol), the ten-year survival was 92% [5]. In our predominantly Afro-descendant population, we reported five- and ten-year renal survival of 93.2 and 82.3%, respectively; overall survival was 94.9% at 10 years, similar to that observed in Caucasians.

Many initial risk factors influencing the outcomes of SLE and LN have been described and highlighted: non-modifiable risk factors such as sex [30,31,32], age at LN onset [3,33], early LN [34], and LN recurrence, clinical and biological parameters such as hypertension [35], elevated initial serum creatinine [5,33,36,37,38], elevated initial proteinuria [33,36,39], pathological information [33,40], chronicity index [41,42,43], and ISN/RPS class [44,45], and genetic factors such as Apol1 [46].

The role of some of these non-modifiable risk factors is still debated, and race or ethnicity is not considered by some authors to be a risk factor for poor outcomes in SLE, in contrast to many modifiable factors including socioeconomic conditions [16,47,48,49,50]. Thus, these non-modifiable parameters at presentation can likely be less unfavorable with better management of modifiable factors.

Some of these modifiable factors are of particular importance among minorities: suboptimal access to care related to distance from the rheumatologist and nephrologist, unaffordable consultation due to insufficient medical insurance [51], access to specialized SLE and LN care [52], ideally organized in a network [40] and without delay in LN diagnosis [33,41], and access to renal biopsy [42]. Additional factors include the choice and dose of induction and maintenance therapies [34,53,54], the availability, prescription, and affordability of immunosuppressants [9,55], the timeliness of medication prescription after diagnosis of LN [33], the use of antiproteinuric and anti-malarial medicine [47,56], non-adherence [22,57], smoking [58], and socioeconomic factors [9,10,19,23,47,57,59].

In Martinique, numerous non-modifiable factors were present in SLE and LN patients, and they shared high-disease activity with other Afro-descendant groups [1]. Nevertheless, many modifiable factors are favorable, and poverty is not an obstacle to care. Transportation is covered by public insurance if necessary to the University Hospital with a Lupus Clinic, labeled by the French Ministry of Health since 2008, included in the national network of rare diseases. All medications are free of charge and there is easy and free access to renal biopsies (which are read twice, locally and in a second specialized center in Paris) and hospitalizations. It should be noted that in Curaçao, less than 10% of LNs were biopsied [60] and that in Barbados, renal biopsy was performed in only 28% of LNs due to lack of resources [61]. Thus, in Martinique, access to SLE specialists is easy and the combination with the proximity of their subspecialists facilitates early diagnosis and optimal treatment of LN patients.

Persistent negative factors can be corrected: adequate induction and maintenance immunosuppressive drugs and generalization of anti-hypertensive drugs and RAAS for better control of hypertension and proteinuria [5,34,35]. It should be balanced that some data transcribed obsolete management practices of different departments (internal medicine, rheumatology, and nephrology departments), before the establishment of the current common strategies. As in numerous studies, our patients with worse outcomes had higher serum creatinine levels at the onset of LN. We did not find any risk factors of severe evolution among the chronicity index or ISN-RPS class but this was probably related to a lack of power.

Regarding treatment, mycophenolate seemed to be associated with better remission rates in Afro-descendants, but no definitive proof has been provided [62]. A low-dose IV CYC regimen but with a high initial dose of steroids has been considered to have the same efficacy in African-American patients than European patients in the ACCESS trial [5,63,64]. This regimen has been systematically proposed. Anti-malarial drugs, associated with better survival, was received by three-quarter of our patients [56]. Adherence to treatment, which remains a concern in the treatment of LN, especially in minorities, is largely underestimated in studies [55,65,66,67]. In our work, all patients were included, treatment compliant or non-compliant, without prior selection as in prospective interventional studies. Although assessed in an unsystematic, self-reported manner, poor adherence to treatment affected one-quarter of our patients. Prescribed to nearly 100% of patients, HCQ was taken for at least 2 consecutive years by 74% of them, mainly due to poor adherence. While data on non-compliance from real-world studies such as ours may be less conclusive than those from controlled studies, none of the latter provide a systematic survey of adherence after treatment initiation. Some limitations of the study should be noted. The number of patients was small, and the retrospective nature of the study did expose it to the risk of missing data. Although vital status or ESRD diagnosis were known for all patients at the end of the study, some patients were lost to follow-up at certain times. Finally, the data collection was old but reflects the reality of the sometimes obsolete or inadequate practices of the different departments before the constitution of a national reference center.

Nevertheless, our data shade the burden of race or ethnicity and highlighted the impact of modifiable factors and access to care in minorities with lupus nephritis. After three decades of improvement, LN survival has not improved since the 2000s, and this may be related to less effective management in minorities living in low-income areas. There is a need for further studies to confirm these results.

## 5. Conclusions

We reported good long-term global and renal survival in a population-based cohort of Afro-Caribbeans with lupus nephritis, similar to those observed in Caucasian patients, arguing for the weight of socioeconomic and modifiable factors.

## Figures and Tables

**Figure 1 jcm-11-04860-f001:**
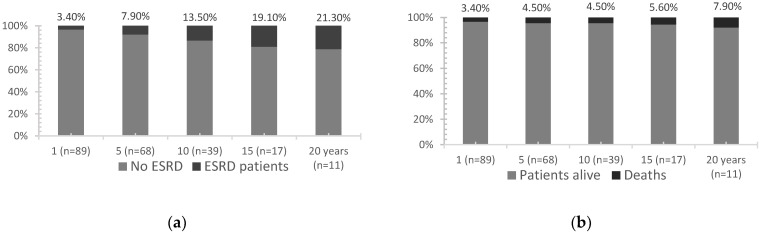
Cumulative rates of ESRD (**a**) and mortality (**b**) during 1, 5, 10, 15, and 20 years in Martinican patients with lupus nephritis.

**Figure 2 jcm-11-04860-f002:**
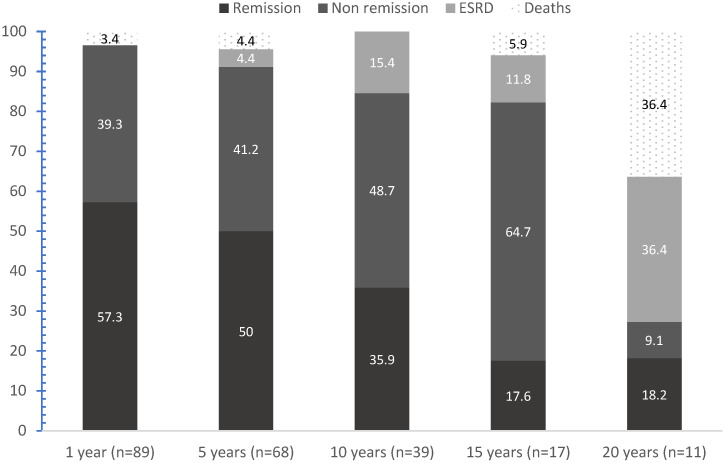
Remission, ESRD, and death rates in Martinican patients with lupus nephritis followed for 1, 5, 10, 15, and 20 years. “(*n* = x)” indicates the remaining patients at the end points. After 1, 5, 10, 15, and 20 years of follow-up, the number of patients lost to follow-up was 2, 4, 7, 4, 2, and 0, respectively. At 1, 5, 10, 15, and 20 years, the number of patients in CR and PR were 38 and 18, 26 and 8, 10 and 4, 1 and 3, and 1 and 1, respectively. Eight patients died: three of infectious origin, one of hemorrhage following abdominal surgery, one of probable massive pulmonary embolism, one of heart failure, and two of unknown cause.

**Figure 3 jcm-11-04860-f003:**
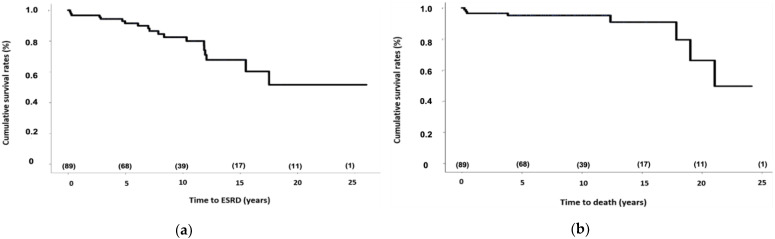
Kaplan Meier analysis of the probability of end-stage renal disease (ESRD) (**a**) or death (**b**) in Martinican patients with lupus nephritis. The numbers in brackets refer to the number of remaining patients. Estimated renal survival at 5, 10, 15, and 20 years were 93.2%, 82.3%, 68%, and 51.7%, respectively. Estimated vital survival at 5, 10, 15, and 20 years were 94.9%, 94.9%, 91%, and 66.7%, respectively.

**Table 1 jcm-11-04860-t001:** Clinical, biological, and histological characteristics at lupus nephritis onset.

	Total (*n* = 89)
**Age,** mean (years)	32.5 ± 13
**Women,** % (*n*)	93.2 (83)
**Duration of SLE before LN (months)**	39.7 ± 60.8
**LN follow-up (months),** mean ± SD	119 ± 72.9
**Clinical features** % (*n*)	
Fever	20.22 (18)
Neurolupus	8.99 (8)
Arthritis	44.94 (40)
Myositis	4.49 (4)
Cutaneous rash	20.22 (18)
Alopecia	10.11 (9)
Mucosal ulcer	2.24 (2)
Serositis	29.21 (26)
SLEDAI, mean	17.11 ± 5.73
**Biological features** % (*n*)	
Anti-ds-DNA Ab	97.75 (87)
Anti-ds-DNA Ab title, mean ± SD	241.34 ± 235.40
Anti-Sm	58.42 (52)
Anti-SSA	56.18 (50)
Anti-SSB	23.59 (21)
Anti-RNP	60.67 (54)
aPL positivity	58.42 (52)
APS	17.98 (16)
Hematuria	73 (65)
Leucocyturia	58.42 (52)
Proteinuria, mean, g/24 h	3.55 ± 3.72
Serum level albumine, mean, g/L	25.19 ± 8.17
Serum level creatinine, mean, µmol/L	118.94 ± 93.21
Low C3	57.3 (51)
Low C4	60.67 (54)
Thrombopenia	5.61 (5)
Leucopenia	12.35 (11)
**Histologic features at first renal biopsy (ISN/RPS),** % (*n*)	
Class I	3.37 (3)
Class II	1.12 (1)
Class III	19.10 (17)
Class IV	24.72 (22)
Class V	19.10 (17)
Class III + V	20.22 (18)
Class IV + V	12.36 (11)
Proliferative LN (III, III + V, IV or IV + V), % (*n*)	76.4 (68)
Activity index, mean, %	35.93 ± 28.8 (36)
Chronicity index, mean, %	23.1 ± 20.6 (36)

Abbreviations: Ab: antibody; aPL: anti-phospholipid auto-antibody positivity; APS: anti-phospholipid syndrome; ISN/RPS: International Society of Nephrology/Renal Pathology Society; LN: lupus nephritis; anti-RNP: anti-ribonucleoprotein auto-antibody; SLE: systemic lupus erythematosus; anti-Sm: anti-Smith auto-antibody; anti SSA: anti-Sjögren’s-syndrome-related antigen A auto-antibody; and anti-SSB: anti-Sjögren’s-syndrome-related antigen B auto-antibodies.

## Data Availability

The data presented in this study are available on request from the corresponding author.

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
