# Peer review of "Good Long-Term Prognosis of Lupus Nephritis in the High-Income Afro-Caribbean Population of Martinique with Free Access to Healthcare"

_jcm, 2022, doi:10.3390/jcm11164860_

Round 1

Reviewer 1 Report

Retrospective analysis of relatively small cohort of 89 patients with lupus nephritis diagnosed and treated within relatively long period (2002-2015) in one renal centre in Martinique suggesting that in high-income country the renal outcome and survival of high-risk Afro-Caribbean patients is good and comparable to Caucasian patients and that the socioeconomic factors may be more important that the ethnic ones. Although the paper is of some interest there are many major and minor comments which are to be taken into consideration. If it is not possible to provide the required data the paper could be also switched to letter to the editor

General comments:

  1. English definitely warrants attention of native speakers – some sentences are incorrect and sometimes it is not easy to undestand what was really meant (some, but not all, examples see below)
  2. Introduction is short, but clearly defining the aim of the study
  3. Discussion is well written, it would be interesting to provide some data on the outcome of (suspected) LN from neighbouring mentioned island, e.g. Barbados or Curacao

Minor comments:

  1. Why urinary creatinine is mentioned among collected parameters in patients with lupus nephritis (Methods)?
  2. Definitions - uncontrolled blood pressure was defined as BP > 130/80 mmHg during at 2 consecutive visits – isn´t it to strict? It may not be easy to achieve in some patients with otherwise very good outcome
  3. Definitions – „LN was suspected in the presence of proteinuria > 0.5g/24h or a urine protein/creatinine ratio > 0.5g/g on two consecutive samples, with or without hematuria (>10000 60 RBCs/mL)“ – it may seem to be clear, but it should be stated that LN was suspected in patients fulfilling the ACR diagnosis of SLE, or otherwise it could be declared that these parameters were used as indication for renal biopsy in all pts (with or without SLE)
  4. „Duration of flare was defined by the time from LN diagnosis to first complete or partial remission, with or without treatment“ – shouldn´t it be better to speak about time to complete or partial remission? I would not designate the activity at time of biopsy as „flare“, the term should be reserved for the new activity after previously achieved remission
  5. „Immunosuppressive treatment for LN was considered at the first administration of any dedicated therapy. Other medications were considered if they have been taken for at least 12 months.“ These statement should be reworded to make sense. Possibly it is meant that patients were treated with IST immediately after diagnosis of LN (some of them were possibly, however, on IST already at time of renal biopsy because of activity of SLE – weren´t they?). I cannot undestand what is meant by other medications and why they should be considered only if taken for at least 12 months
  6. Line 72 – instead of „creatinemia“ I would definitely prefer serum level of creatinine
  7. Line 73 – instead of „urinal samples“ should be urinary samples
  8. Line 74 - „Relapse duration was considered until remission“, I would speak about re-remission.
  9. Table 1 – there should be serositis instead of „seritis“
  10. Table 1 – instead „Ds DNA Ab title“ should be – anti-ds-DNA Ab titre
  11. Table 1 – once again instead of „creatinemia“ there should be at least creatininemia
  12. What was the proportion of patient with impaired renal function (CKD3a, 3b and 4, or 5) at presentation?L
  13. Line 96 – „Although the overall time to loss to follow-up was 4.2% of the total follow-up time, the renal and vital status of every single patient was known at the end of the study“ – how may patients were lost from FU? Does it mean that most of those lost from FU were lost just after diagnosis? It is quite surprising that data from the end of FU were finally obtained for all lost patients – we were never so successful in this kind of retrospective surveys
  14. Line 98 – it is not clear if the authors speak about all patient with lupus nephritis, or only those with proliferative lupus nephritis
  15. It is also not clear why HCQ was used for two years by 74% only – the rest did not tolerate the treatment, or does it reflect poor adherence to treatment?
  16. What was the indication of cyclophosphamide, or mycophenolate mofetil, e.g. were the patient with more severe disease treated with cyclophosphamide? What were the doses of CPH (NIH vs. Euro lupus regimen?)
  17. Why were 20 patients treated (in one centre) inadequately?
  18. Was the treatment policy changes between 2002 – 2015? I suppose so and if yes, it should be clearly stated and there should be subanalysis of pts treated before and after the change of the treatment policy
  19. There should be also some explanation for very heterogenous maintenance therapy (in one centre)
  20. Line 106 – those 13 patients with transformation from LN V to proliferative LN must have been re-biopsied. What was the indication for rebiopsy? Were these patients also included among those with proliferative LN?
  21. Line 111 – time to remission (18 months) is very long. Any explanation for that?
  22. Line 127 – what was the propotion of patients with „uncontrolled hypertension“ and how were the patients treated? Was the main reason of „uncontrolled hypertension poor adherence, or real resistence to treatment?
  23. What was the outcome (in terms of survival) of patients reaching ESRD?
  24. Line 189 – did the authors compared the outcome of patients treated initially with CPH and MMF?
  25. Was there any relation between achieved proteinuria and longterm outcome (as in MAINTAIN trial in the cited paper of Tamirou et al.)?
  26. Most references are quite old and should be supplemented with some recent ones, e.g. Mackay et al., 2019

Author Response

Good morning

Reviewer 2 Report

The present work aimed to present the outcomes of lupus nephritis (LN) in Afro-Caribbean patients with free access to healthcare. The subject is interesting and relevant to both clinicians and researchers, especially seeing as socioeconomic issues are less often taken into account when analyzing racial differences in systemic lupus erythematosus (SLE) and other autoimmune diseases. Apart from free specialist consultations, free biopsies and medication, the LN cohort analyzed by the authors also had the advantage of free transportation to receive medical care.

The manuscript is rather well organized. Nevertheless, certain issues should be addressed by the authors:

  1. The mean follow-up of patients with LN was ~10 years (the minimum was 12 months, while the maximum follow-up period was not mentioned), yet the analysis of ESRD and mortality included outcomes at 15 and 20 years. Was this analysis performed according to the duration of SLE or specifically LN? The duration of SLE is not mentioned in Table 1. Please add the missing data (including SLE duration).
  2. The patients included in the study had a follow-up period of 12 months minimum (except for the participants who died earlier than 12 months after their inclusion in the study). However, the mortality rates were evaluated up to 20 years (hence, the follow-up period was between 1-20 years). Therefore, the follow-up period probably varied greatly between patients. Please present all outcome measures separately at 1 year, at 5 years, at 10 years, at 15 years, and at 20 years (detailed - remission, flares, mortality rates and causes of death, ESRD, CR, PR), with the mention of the final number of patients in the subgroups (patients followed for at least 5, at least 10, at least 15, up to 20 years). Moreover, the number of patients who were lost from follow-up should be mentioned at 5 years, 10 years, 15 years, and 20 years. The final subgroups – at 5/10/15/20 years should be updated accordingly, taking into account the number of patients lost from follow-up, as well as previous deaths.
  3. Please provide more details regarding treatment at 1/5/10/15 years and how it may have influenced LN outcomes in your cohort.
  4. Please provide more data on comorbidities and how they may have influenced LN outcomes.
  5. Please update the statistical analysis and the discussions according to the newly presented data (Points 2-4). In your discussions, please search for specific chronology (outcomes at 1/5/10/15/20 years) regarding LN outcomes in Caucasian versus non-Caucasian patients (or specifically in patients of African and/or Afro-Caribbean descent), where available.
  6. The limitations of the study are not described clearly in the discussions (ex: the low number of patients, the retrospective nature of the study, different follow-up periods, the lack of a control group, etc.).
  7. Please define each acronym before using it in the text.
  8. Please describe the acronyms used in tables and figures (the explanation may be added beneath the tables/figures).
  9. The title of the manuscript mentions “high income”, yet the results do not include the analysis of the patients’ income. Please add the missing data.

Author Response

Good morning, 

Round 2

Reviewer 2 Report

Dear Authors,

Thank you for responding to the comments in such a detailed manner, I believe that the changes made to the work provide more clarity and further highlight the importance of your work.

Beginning with line 169 of the revised manuscript), please add more details pertaining to the long-term outcomes of Caucasian patients with similar access to healthcare as your own study group (including specific percentages - ex: mortality rates at 10, 15 and 20 years of follow-up). A phrase comparing this information to your own results would further support the statement: "we report good long-term outcomes of our Afro-Caribbean population, similar to that observed in Caucasian patients" (lines 168-170).

At this stage, I recommend acceptance after the minor changes mentioned above.

Author Response

Good morning 

And thank you for your time

We have added the following information according to your recommendations (line 173 of the revised manuscript):  

" Indeed, some studies have compared the long-term outcomes of lupus nephritis between patients of African origin and Caucasians. For example, 1, 5, and 10 years-survival of African descent versus Caucasians LN patients was 95 vs 94%, 71 vs 85%, and 59 vs 81%. At the same end points, renal survival was 85 vs 91%, 50 vs 74%, and 38 vs 68% [4]. Another study reported a 5-year renal survival in African-descendants of 57% compared to 94.5% in Caucasians [27]. In a largely Caucasian population (76 Caucasians, 8 Afro-descendants, 6 Asians), who had received recent immunosuppressive treatments (Eurolupus protocol), the 10-year survival was 92% [5]. In our predominantly Afro-descendant population, we report 5- and 10-year renal survival of 93.2 and 82.3%; overall survival was 94.9% at 10 years, similar to those observed in Caucasians.